# S-nitrosylation of the zinc finger protein SRG1 regulates plant immunity

Beimi Cui[1,2,3], Qiaona Pan[1,2,3], David Clarke[4], Marisol Ochoa Villarreal[3], Saima Umbreen[3], Bo Yuan[1,2], Weixing Shan[5], Jihong Jiang[1,2] & Gary J. Loake [1,3]

Nitric oxide (NO) orchestrates a plethora of incongruent plant immune responses, including the reprograming of global gene expression. However, the cognate molecular mechanisms remain largely unknown. Here we show a zinc finger transcription factor (ZF-TF), SRG1, is a central target of NO bioactivity during plant immunity, where it functions as a positive regulator. NO accumulation promotes *SRG1* expression and subsequently SRG1 occupies a repeated canonical sequence within target promoters. An EAR domain enables SRG1 to recruit the corepressor TOPLESS, suppressing target gene expression. Sustained NO synthesis drives SRG1 S-nitrosylation predominantly at Cys87, relieving both SRG1 DNA binding and transcriptional repression activity. Accordingly, mutation of Cys87 compromises NO-mediated control of SRG1-dependent transcriptional suppression. Thus, the SRG1-SNO formation may contribute to a negative feedback loop that attenuates the plant immune response. SRG1 Cys87 is evolutionary conserved and thus may be a target for redox regulation of ZF-TF function across phylogenetic kingdoms.

[1] Jiangsu Normal University - Edinburgh University, Centre for Transformative Biotechnology of Medicinal and Food Plants, Jiangsu Normal University, 101 Shanghai Road, Xuzhou, P.R. China. [2] Key Laboratory of Biotechnology for Medicinal Plants, Jiangsu Normal University, 101 Shanghai Road, Xuzhou, P.R. China. [3] Institute of Molecular Plant Sciences, University of Edinburgh, Edinburgh EH9 3BF, UK. [4] School of Chemistry, University of Edinburgh, Edinburgh EH9 3FJ, UK. [5] State Key Laboratory of Crop Stress Biology for Arid Areas and College of Agronomy, Northwest A&F University, Yangling, Shaanxi 712100, China. These authors contributed equally: Beimi Cui, Qiaona Pan. Correspondence and requests for materials should be addressed to G.J.L. (email: gloake@ed.ac.uk)

A key feature following pathogen recognition in eukaryotes is the engagement of a nitrosative burst, leading to the accumulation of the gaseous signalling molecule, nitric oxide (NO)[1,2]. In parallel, there is a rapid synthesis of reactive oxygen intermediates (ROIs)[3,4]. These small, redox-active molecules orchestrate a plethora of immune responses in plants including cell wall structural protein cross-linking[5], salicylic acid (SA) synthesis[6–8] and signalling and pathogen-triggered, programmed cell death development[9–11].

S-nitrosylation, the addition of an NO moiety to a protein cysteine (Cys) thiol to form an S-nitrosothiol, has emerged as a key mechanism for the transfer of NO bioactivity[2,6,12]. However, only rare, highly reactive, solvent exposed Cys thiols, often embedded within an SNO motif, are potential sites for this redox-based, post-translational modification[13–15]. SNO formation at target Cys thiols can regulate protein function acting akin to other more established molecular switches such as phosphorylation[16]. The total cellular level of S-nitrosylation is controlled indirectly by the action of the enzyme S-nitrosoglutathione reductase (GSNOR), which turns over the natural NO donor, S-nitrosoglutathione (GSNO)[6,17,18]. This enzyme is required for plant development in addition to biotic and abiotic responses[17,19,20]. Significantly, recent evidence implies that GSNO and NO may have separable and overlapping functions integral to redox regulation, implying distinct reactive nitrogen species (RNS) may have discreet biological activities[21,22].

It is well established that NO accrual following the pathogen-triggered nitrosative burst contributes to the reprogramming of broad suites of defence-related genes during plant immune function[23–27]. However, the molecular mechanism(s) responsible remain largely undetermined. To date, NO has been proposed to control the translocation of the transcriptional co-activator NPR1 into the nucleus[7,28] and the specific DNA-binding activity of its protein interactor, the basic leucine-zipper transcription factor, TGA1, which regulates the expression of Pathogenesis Related (PR) genes[28].

Here we show that expression of the zinc finger transcription factor, SRG1, is induced following the pathogen-triggered nitrosative burst. Subsequently, this TF binds to either ACTN$_6$ACT or ACTN$_4$ACT sequences in target genes that presumably include negative regulators of immune function. SRG1 appears to act as a transcriptional repressor utilizing its putative ERF-associated amphiphilic repression (EAR) domain to recruit the corepressor TOPLESS, contributing to the engagement of plant defence responses and the establishment of immunity. As NO accumulates SRG1 becomes S-nitrosylated, with Cys87 a major target, this disables zinc coordination, abolishing SRG1 DNA binding and transcriptional repression activity. The absence of SRG1 occupancy at target promoter sites may subsequently release repression by negative regulators, contributing to the cessation of transient immune responses.

## Results

### NO regulates SRG1 expression.
A key feature of NO function during plant immunity is thought to be the regulation of specific sets of defence-related genes[9,10,23–27]. However, the molecular mechanism(s) underpinning the control of these gene networks remain to be established. Following the interrogation of both public and in-house data sets[24,27], derived from the profiling of Arabidopsis gene expression in response to (S)NO accumulation, we identified a C2H2 type zinc finger transcription factor (ZF-TF) of the C1-2i subclass, encoded by At3g46080, consisting of two zinc fingers with a QALGGH sequence, a conserved feature of the zinc finger (ZF) domain[29]. ZF-TFs are one of most prevalent regulatory proteins amongst eukaryotes[30]. This motif consists of

approximately 30 amino acids with two pairs of conserved Cys and His residues binding tetrahedrally to a zinc ion[31]. Transcripts corresponding to the identified Zn-TF were rapidly induced in response to (S)NO. We designated this transcription factor (TF) SNO-regulated gene1 (SRG1) (Supplementary Figure 1a), a previously uncharacterised member of the ZF of Arabidopsis (ZAT) gene family[32,33]. ZF-TFs comprise a large, 176 member family of TFs in Arabidopsis[29] and proteins of this class have previously been linked with stress responses[33,34] and plant development[35].

To confirm the impact of (S)NO on SRG1 expression, we carried out qRT-PCR analysis, enabling quantification of SRG1 expression in response to exogenous application of the NO donor, sodium nitroprusside (SNP)[20], in the presence or absence of the NO scavenger, 2-4-carboxyphenyl-4,4,5,5-tetramethylimidazoline-1-oxyl-3-oxide (cPTIO)[9]. SNP induced SRG1 expression at 6 h and more strongly by 24 h post application (Fig. 1a). Furthermore, transgenic plants possessing a SRG1 promoter fused to the β-glucoronidase (GUS) reporter gene, SRG1::GUS, exhibited GUS activity following exogenous SNP application and in the presence of cPTIO, the GUS activity of these plants was significantly reduced (Supplementary Figure 1b). The observed changes in GUS activity following SNP treatment were also quantified in either the presence or absence of cPTIO (Fig. 1b).

To determine if SRG1 was pathogen responsive, we examined SRG1 transcript levels by qRT-PCR in wild-type Arabidopsis Col-0 plants in response to virulent Pseudomonas syringae pv. tomato DC3000[36] or Pst DC3000 expressing the avirulence (avr) gene avrRpm1[37]. In this context, the avr gene product is recognised by the RPM1 Resistance (R) protein in the Col-0 accession of Arabidopsis[37]. SRG1 expression was induced by both Pst DC3000 and Pst DC3000(avrRpm1) at 3 h and 6 h post infiltration (hpi). However, SRG1 was induced more strongly following RPM1-mediated pathogen recognition (Fig. 1c). Similar results were obtained when SRG1::GUS transgenic plants were challenged with these two bacterial strains and GUS activity assayed (Supplementary Figure 1c). The observed changes in GUS activity were also quantified (Fig. 1d). Finally, we compared SRG1 expression in atgsnor1-3 plants, which exhibit higher SNO levels compared to wild-type following challenge with Pst DC3000(avrRpm1)[6]. SRG1 expression was enhanced and accelerated in atgsnor1-3 plants relative to the wild-type line (Supplementary Figure 1d). In contrast, the SA marker gene, Pathogenesis Related 1 (PR1), was induced by 6 hpi in wild-type plants but the induction of this gene was significantly reduced at 6 hpi in atgsnor1-3 plants, which are compromised in SA synthesis and signalling[6,7] (Supplementary Figure 1d).

To examine the subcellular localisation of this TF, SRG1 was fused with green fluorescent protein (GFP) and transiently expressed in Arabidopsis protoplasts (Fig. 1e) or within Nicotiana benthamiana leaves mediated by Agrobacterium tumefaciens GV3101 (Supplementary Figure 1e). As anticipated, SRG1 localized to the nucleus, whereas free GFP was observed throughout the cell (Fig. 1e and Supplementary Figure 1e). To confirm this localization pattern, a nuclear localisation sequence (NLS) from the SV-40 T antigen was fused to GFP (NLS-GFP)[38] and utillized as a control along with 4′,6-diamidino-2-phenylindole (DAPI) staining for nuclear localisation (Supplementary Figure 1f). In aggregate, this data implied SRG1 was localised to the nucleus.

Collectively, these results suggest that SRG1 maybe transcriptionally activated in response to NO accumulation either driven chemically or following engagement of the pathogen-triggered nitrosative burst. Further, SRG1 induction appears to be independent of SA synthesis and signalling. Our data also implies SRG1 may localise to the nucleus.

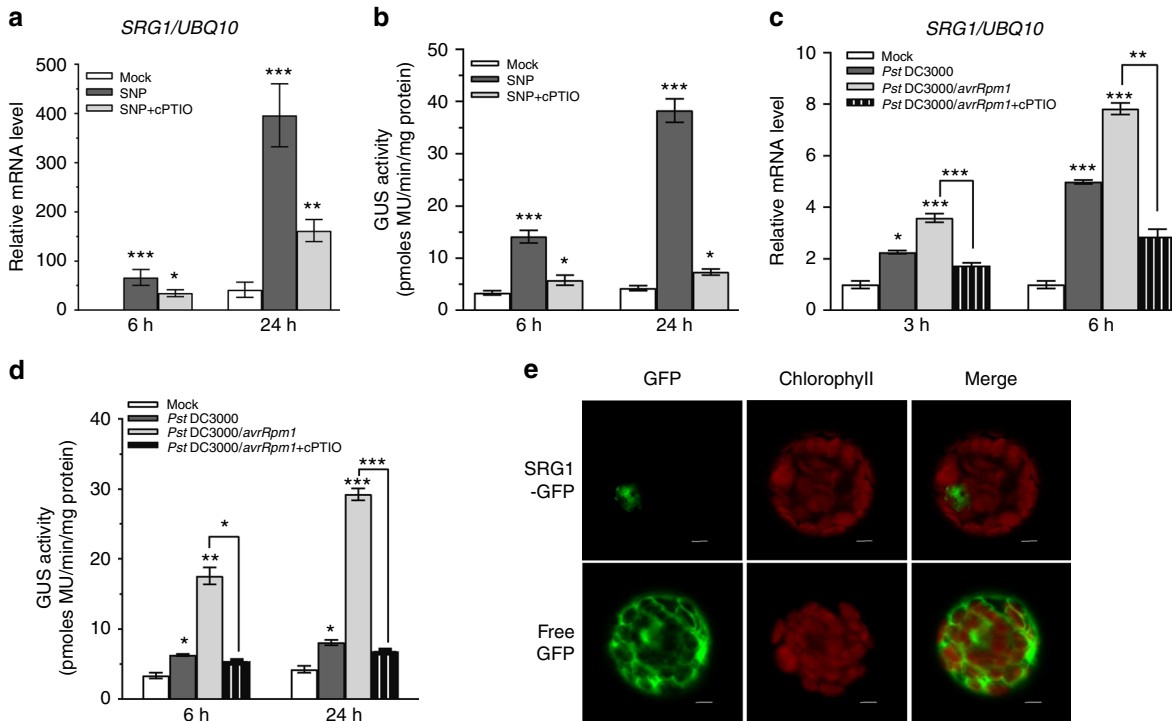

**Fig. 1** *SRG1* expression is regulated by nitric oxide. **a** Transcript levels of *SRG1* were determined following treatment with the nitric oxide (NO) donor, sodium nitroprusside (SNP), either alone or in combination with the NO scavenger 2-4-carboxyphenyl-4,4,5,5-tetramethylimidazoline-1-oxyl-3-oxide (cPTIO). **b** β-glucuronidase (GUS) activity upon SNP treatment of *SRG1::GUS* lines. GUS activity in the given *SRG1::GUS* line in response to SNP (300 μM) and SNP plus cPTIO (200 μM) was analyzed by a GUS activity assay. **c** Quantitative real time polymerase chain reaction (qRT-PCR) to quantify *SRG1* gene expression post treatment with *Pseudomonas syringae* pv *tomato* DC3000 or *Pst* DC3000 expressing *avrRpm1* in the presence or absence of cPTIO. **d** GUS activity assay of *SRG1::GUS* lines post treatment with either *Pst* DC3000 or *Pst* DC3000(*avrRpm1*) in the presence or absence of cPTIO. **e** Subcellular localization of SRG1-green fluorescent protein (GFP) and free GFP were analysed in *Arabidopsis* protoplasts using a confocal microscope. Scale bar, 5 μm. Error bars represent mean ± standard deviation (SD) ($n = 3$ independent experiments). Asterisks indicate a significant difference from mock (student *t*-test, ***$P < 0.001$, *$P < 0.05$)

**SRG1 is a positive regulator of plant immunity**. To examine the contribution of *SRG1* to plant immune function, we generated *SRG1* overexpression and T-DNA loss-of-function lines. Plants containing a Cauliflower Mosaic Virus 35S (CaMV35S)::*SRG1* transgene exhibited reduced stature. In contrast, an *srg1* T-DNA insertion mutant that exhibited almost no detectable *SRG1* expression (Supplementary Figure 2a) was larger than wild-type Col-0 plants (Fig. 2a). Furthermore, the fresh weight of these lines directly correlated with the strength of *SRG1* expression (Figs 2b, c). Thus, indicating *SRG1* negatively impacts *Arabidopsis* stature.

To explore the potential impact of *SRG1* on basal resistance, the generated lines were challenged with *Pst* DC3000. Interestingly, loss-of-function *srg1* plants exhibited enhanced disease susceptibility towards this pathogen, whereas CaMV35S::*SRG1* lines showed enhanced resistance (Fig. 2d). To explore if *SRG1* also affected *R* gene-mediated resistance, the same lines were challenged with *Pst* DC3000(*avrRpm1*). While *srg1* plants supported an increased titre of *Pst* DC3000(*avrRpm1*) relative to wild-type plants, in CaMV35S::*SRG1* lines the titre of this pathogen was reduced (Fig. 2e). To confirm the increased disease susceptibility observed in *srg1* plants was due to loss of *SRG1* function we complemented *srg1* lines with a wild-type copy of *SRG1*. The complemented *srg1* line restored bacterial titres to those supported in wild-type plants (Supplementary Figure 2b-d). These results imply that *SRG1* is a positive regulator of both basal defence and *R* gene-mediated resistance.

To determine the molecular basis of enhanced resistance in CaMV35S::*SRG1* lines we quantified *PR1* expression by

qRT-PCR. The expression of this gene was increased in CaMV35S::*SRG1* plants relative to wild-type and the magnitude of expression directly correlated with the abundance of *SRG1* transcripts (Figs 2c, f). Further, CaMV35S::*SRG1* lines exhibited increased levels of the immune activator, SA (Fig. 2g). Conversely, *PR1* expression was reduced in *srg1* plants (Supplementary Figure 2e). 3,3′-diaminobenzidine (DAB) and nitro blue tetrazolium (NBT) staining revealed that the ROIs hydrogen peroxide ($H_2O_2$) and superoxide ($O_2^{·−}$), respectively, also accumulated to higher levels in CaMV35S::*SRG1* plants relative to wild-type in the absence of attempted pathogen infection (Figs 2h, i and Supplementary Figure 2f and 1g). Similar results were obtained when we quantified the intensity of DAB and NBT staining (Supplementary Figure 2h and 2i). Cell death development in these lines was also determined by trypan blue (TB) staining, revealing that some CaMV35S::*SRG1* plants exhibited micro lesions (Fig. 2j). Quantification of TB intensity staining also supported the observation that overexpression of *SRG1* elevated cell death development (Supplementary Figure 2j). Further, we also quantified cell death development by electrolyte leakage following challenge with *Pst* DC3000(*avrRpm1*). Cell death development was decreased in *srg1* plants relative to wild-type and increased in CaMV35S::*SRG1* lines (Fig. 2k). Taken together, these data suggest that *SRG1* is a positive regulator of cell death development, SA synthesis and signalling, ROI accumulation and disease resistance.

**S-nitrosylation of SRG1 represses its DNA-binding activity.** The emerging evidence from animal and microbial systems

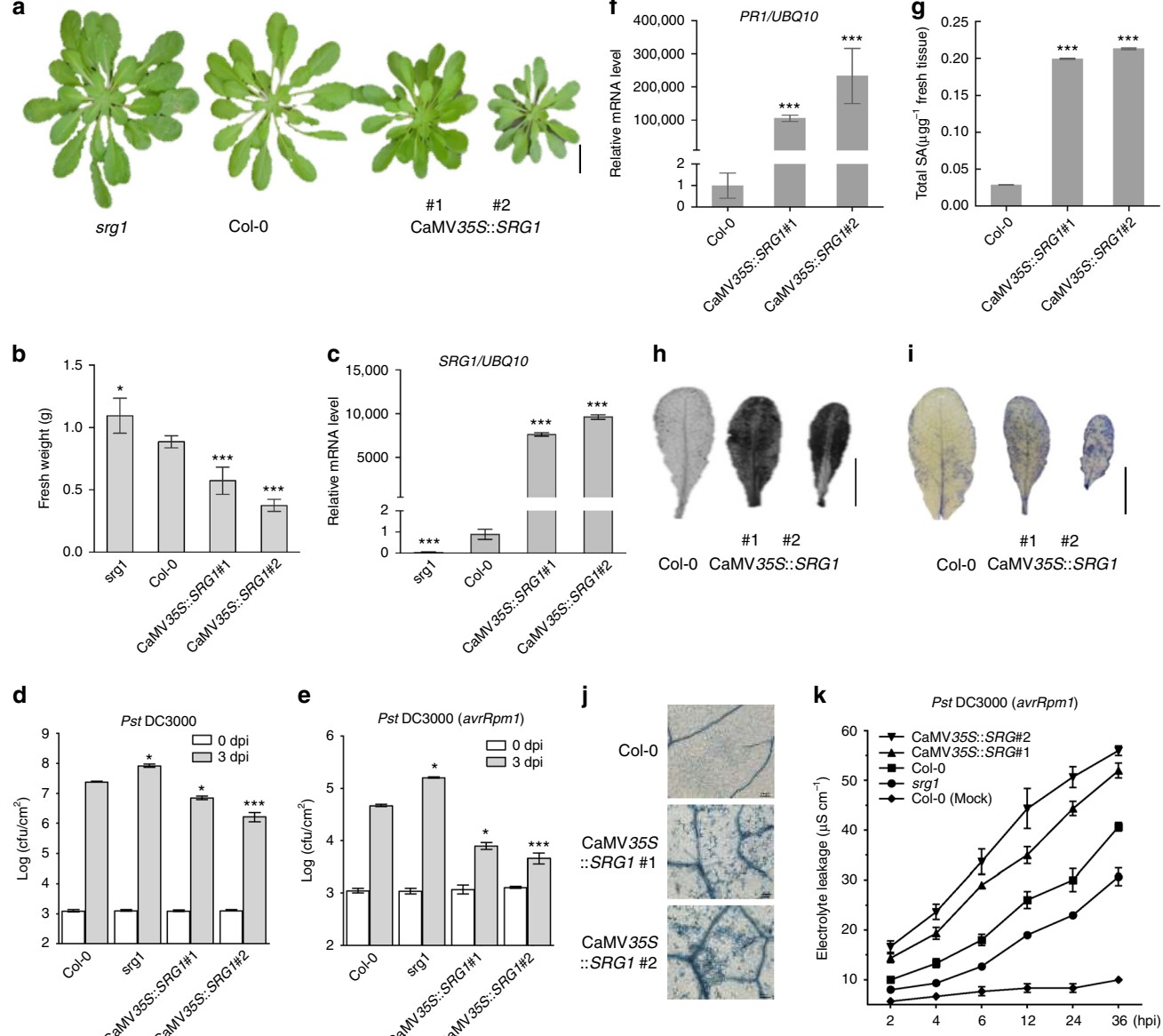

**Fig. 2** SRG1 is a positive regulator of plant immunity. **a** Morphological phenotype of CaMV35S::SRG1 overexpression lines. The indicated lines were soil-grown under short day conditions and photographed at 6-weeks-old. Scale bar, 2 cm. **b, c** The fresh weight (**b**) and mRNA level of SRG1 (**c**) in the stated Arabidopsis lines. Error bars represent ± SD from 3–6 biological replicates (t-test, ***P < 0.001, *P < 0.05, significantly different from Col-0). **d, e** Titre of Pst DC3000 (**d**) and Pst DC3000(avrRpm1) (**e**) in the indicated plant genotypes at 0 days post infiltration (dpi) and 3 dpi. Error bars represent ± SD from 6 biological replicates and student t-test shows significant difference from Col-0 (***P < 0.001, *P < 0.05). **f** Pathogenesis-related 1 (PR1) gene expression determined by quantitative real-time polymerase chain reaction (qRT-PCR) in the indicated plant lines. Error bars indicate ± SD (n = 3 and ***P < 0.001 by t-test compared with Col-0). **g** Total salicylic acid (SA) levels in the given plant genotypes. Error bars indicate ± SD (n = 3 and ***P < 0.001 by t- test compared with Col-0). **h, i** Accumulation of hydrogen peroxide and superoxide detected by either 3,3'-diaminobenzidine (DAB) (**h**) or nitro blue tetrazolium (NBT) (**i**) staining, respectively, in the given plant lines. Scale bar, 0.5 cm. **j** Cell death development was scored by trypan blue staining. Scale bar, 100 μm. **k** Ion leakage was recorded post Pst DC3000(avrRpm1) challenge at the indicated hours post inoculation (hpi). Error bars represent ± SD from 3 biological replicates. Asterisks indicate a significant difference compared with Col-0 (student t-test, ***P < 0.001, *P < 0.05)

suggests that ZF-TFs might be important targets for redox regulation, however, the mechanistic details remain to be fully established[39,40]. As S-nitrosylation is a key mechanism to convey NO bioactivity, we determined if SRG1 might be a target for this redox-based post-translation modification. Thus, we cloned and expressed SRG1 and purified the cognate recombinant protein, which was subsequently exposed to GSNO at concentrations typically used to score for S-nitrosylation in vitro[8,11]. Possible formation of SRG1-SNO was scored by the biotin-switch technique (BST)[36]. Exposure of SRG1 to the natural NO donor, GSNO, resulted in significant S-nitrosylation of SRG1 (Fig. 3a). Furthermore, the extent of SRG1-SNO formation was directly proportional to the GSNO concentration and the addition of dithiothreitol (DTT) strikingly reduced SNO-SRG1 formation, consistent with the presence of a reversible thiol modification (Fig. 3b).

To determine if SRG1 could be S-nitrosylated in vivo, we generated a FLAG-tagged SRG1 (FLAG-SRG1) gene fusion using a

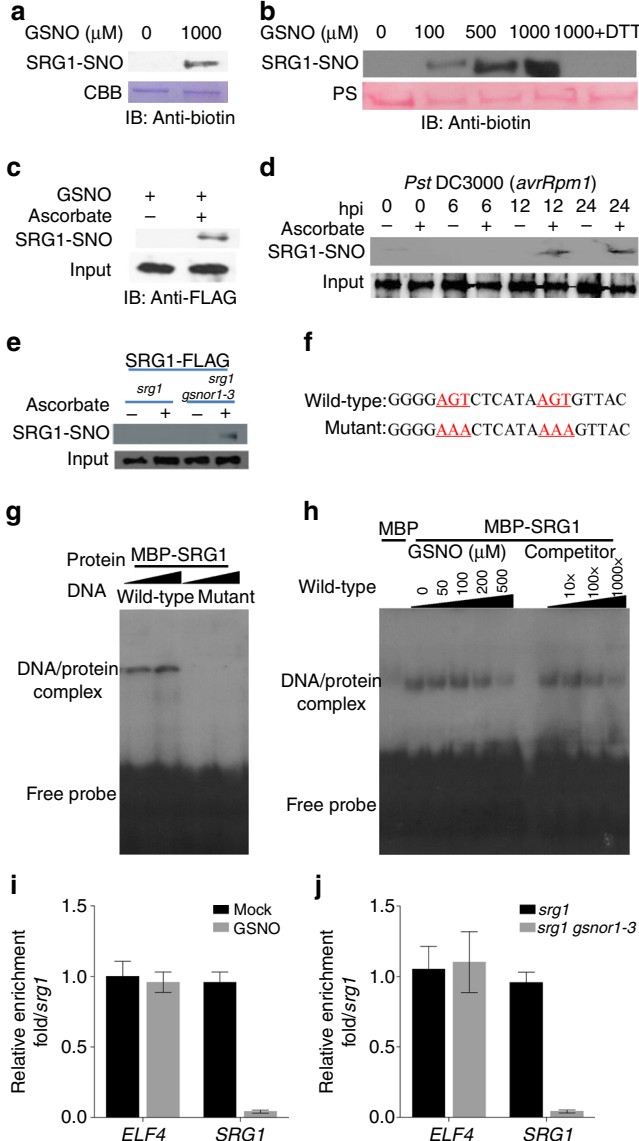

**Fig. 3** Regulation of SRG1 DNA-binding activity by *S*-nitrosylation. **a** *S*-nitrosoglutathione (GSNO)-induced *S*-nitrosylation of SRG1 in vitro. Recombinant SRG1 was subjected to the biotin-switch assay and the resulting protein was interrogated by western blot analysis utilising an anti-biotin antibody. CBB, Coomassie Brilliant Blue protein stain. **b** Recombinant MBP-SRG1 was incubated with the stated concentrations of GSNO and subsequently subjected to the biotin-switch assay. PS, Ponceau S stain. **c** Total protein extracts from *srg1* plants expressing SRG1-FLAG were subjected to the biotin-switch assay after exposure to 1 mM GSNO for 10 min, with or without ascorbate to control for SNO formation. **d** *Pst* DC3000 induced *S*-nitrosylation of SRG1. Total protein extracts from *srg1* plants expressing SRG1-FLAG were subjected to the biotin-switch assay after *Pst* DC3000 inoculation. Ascorbate was employed as indicated to control for SNO formation and the input indicated that the total SRG1-FLAG used for IP. **e** Total protein extracts from either *srg1* or *srg1 gsnor1-3* plants expressing SRG1-FLAG were subjected to the biotin-switch assay. Ascorbate was employed as indicated to control for SNO formation and the input indicated that the total SRG1-FLAG used for IP. **f** Sequence of wild-type synthetic probe and its associated mutant version utilised in **g** and **h**. **g** The indicated wild-type or corresponding mutant oligonucleotide DNA sequences, were labelled with [γ-$^{32}$P]-ATP and the ability of SRG1 to bind to these sequences determined by EMSA. **h** Recombinant SRG1 was incubated with the stated concentrations of GSNO and subsequently subjected to the EMSA assay. All experiments were repeated in triplicate. **i**, **j** ChIP assays scoring the impact of GSNO on SRG1-GFP binding to its cognate promoter containing the binding site indicated in **f**. Wild-type Col-0 protoplasts transiently expressing SRG1-GFP were treated with 1 mM GSNO and subjected to ChIP analysis using an anti-GFP antibody (**i**). Protoplasts from either *srg1* or *srg1 gsnor1-3* plants were subjected to ChIP analysis (**j**). *ELF4* was utilised as an internal control. Also, the mock value in **i** or *srg1* value in **j** were set to 1 after normalization. Error bars represent ± SD from 3 biological replicates

gateway system[41]. Expression of FLAG-SRG1 in *srg1 Arabidopsis* protoplasts could be detected with an anti-FLAG antibody, whereas there was no signal in control plants (Supplementary Figure 3). Subsequently, *Arabidopsis* protoplasts expressing this transgene were exposed to GSNO and endogenous proteins subjected to the BST, subsequently biotinylated proteins were purified with streptavidin beads. These proteins were then immunoblotted with an anti-FLAG antibody. SRG1 was found to be *S*-nitrosylated in vivo (Fig. 3c). We next determined if SRG1 was *S*-nitrosylated during the plant immune response. Following *Pst* DC3000(*avrRpm1*) inoculation, SRG1-SNO formation was detected at 12 and 24 hpi, suggesting that *S*-nitrosylation of SRG1 is promoted during the later stages plant immune function (Fig. 3d), Further, SRG1-SNO formation was increased in protoplasts derived from *srg1 atgsnor1-3* plants relative to *srg1* (Fig. 3e). Collectively, these findings imply that SRG1 is *S*-nitrosylated both in vitro and in vivo. Further, SRG1-SNO formation occurs during the later stages of plant immune function.

To examine the potential biological consequences of SRG1 *S*-nitrosylation, the possible impact of this modification on SRG1 DNA binding was determined. SRG1 is a member of the

C2H2 class of ZF-TFs which bind DNA sequences with a repeated AG/CT motif[42]. Analysis of the *SRG1* promoter revealed an abundance of this motif implying possible auto-regulation. Thus, we assayed DNA sequences from the *SRG1* promoter containing a repeated AG/CT motif as possible sites for SRG1 binding (Supplementary Figure 4a and b and Fig. 3f). SRG1 exhibited binding activity towards two of these DNA sequences that exhibited either a 4 or 6 bp spacing between motifs but not to corresponding sequences in which the AG/CT core motif was mutated (Fig. 3g and Supplementary Figure 4c–h). SRG1 therefore selectively binds AG/CT motifs within its own promoter implying possible auto-regulation. The identified AGTN$_6$AGT and ACTN$_4$ACT binding sequences were found at a frequency of 0.000522288 and 0.00048702, respectively, in the *Arabidopsis* genome sequence, spread relatively evenly across all five chromosomes (Supplementary Tables 3, 4).

As our data suggests that SRG1 is *S*-nitrosylated, we determined if this redox-based post-translational modification might modulate the DNA-binding activity of SRG1. Application of GSNO reduced the amount of SRG1–DNA complex formation in a concentration-dependent fashion, suggesting that *S*-nitrosylation could blunt SRG1 DNA-binding activity in vitro (Fig. 3h). We next examined if the DNA-binding activity of SRG1 was regulated by *S*-nitrosylation in vivo. Chromatin immunoprecipitation (ChIP) analysis was therefore performed to determine SRG1 DNA binding to a sequence incorporating its identified cognate AG/CT core motif. In contrast to eukaryotic initiation factor 4A (EIF4A)[43], where binding to its associated cis-element was unaffected by increased GSNO, SRG1 DNA-binding activity

was strikingly reduced (Fig. 3i). In a similar fashion, while EIF4A binding was not affected in a *gsnor1-3* genetic background, which exhibits increased SNO levels[6], SRG1 binding to its cognate DNA sequence motif was strikingly reduced (Fig. 3j).

Collectively, our findings suggest that NO function might negatively regulate SRG1 DNA-binding activity via *S*-nitrosylation of one or more target Cys residues both in vitro and in vivo.

**SRG1 *S*-nitrosylation reduces SRG1 transcriptional repression.** To explore the biological function of SRG1 in vivo, we tested if this protein might regulate transcription, as our data suggested SRG1 may specifically bind a promoter cis-element. Significantly, SRG1 contains a leucine-rich ETHYLENE RESPONSE FACTOR-associated amphiphilic repression (EAR) motif-like sequence within its C-terminus, defined by the consensus sequence pattern LxLxL (Supplementary Figure 5)[44,45]. The presence of EAR motifs in some plant proteins has been demonstrated to mediate interactions with co-repressors to form a transcriptional repressor complex[46]. Thus, SRG1 was assessed for its possible interaction with TOPLESS, a corepressor which has previously been shown to interact with the EAR motif[47]. The N-terminus of TOPLESS, containing its protein interaction motif, was utilized in a glutathione S-transferase (GST) pull-down assay to test for possible binding to SRG1. This experiment revealed that SRG1 could interact with TOPLESS in vitro (Fig. 4a). A yeast two-hybrid assay confirmed this interaction (Fig. 4b). Further, a bimolecular fluorescence complementation (BiFC) assay suggested this interaction might occur in vivo (Fig. 4c). Collectively, these data imply that SRG1 might recruit the corepressor TOPLESS to form a transcriptional repressor complex at SRG1 DNA-binding sites.

To clarify if SRG1 exhibits transcriptional repression activity in vivo, an *Arabidopsis* transient transcription activity assay was conducted[48]. *SRG1* was fused to the C-terminus of the *galactose 4 DNA binding domain* (*GAL4-BD*) under the control of the CaMV*35S* promoter (*35 s::GAL4-SRG1*) with the resulting protein product assayed for transcriptional activity utilizing a reporter gene comprised of 5 copies of the cognate Galactose 4 (GAL4) DNA-binding site fused to the firefly *Luciferase* reporter gene (Supplementary Figure 6). A plasmid containing *a Renilla LUC* gene driven by the CaMV*35S* promoter was co-transformed as a normalization control (Supplementary Figure 6). SRG transcriptional activity was tested after introducing a given effector plasmid along with the reporter plasmid into *Arabidopsis srg1* protoplasts. LUC activity was reduced by ~50% in the presence of SRG-GAL4 (Fig. 4d). This data implies that SRG1 functions as a transcriptional repressor *in planta*.

We next examined if the EAR-like motif within the C-terminus of SRG1 is required for its transcriptional repression activity (Supplementary Figure 6). When the C-terminus of SRG1 containing the EAR domain was deleted, both its transcriptional repression activity and its interaction with TOPLESS were blocked (Fig 4d, e). We next determined if (S)NO accumulation might affect SRG1 transcriptional repressor function. Application of the NO donor, SNP, strikingly reduced the ability of SRG1 to operate as a transcriptional repressor (Fig. 4f). Similarly, in *srg1 atgsnor1-3* and *srg1 nox1* mutants, which primarily accumulate either SNOs or NO, respectively[6,49], the transcriptional repressor activity of SRG1 was abolished (Fig. 4g). Thus, our data implies that *S*-nitrosylation inhibits the transcriptional repression activity of SRG1.

To examine if the abolition of SRG1 transcriptional repressor activity by (S)NO correlates with increased *S*-nitrosylation of this TF, we determined the level of SNO-SRG1 formation in *Pst* DC3000(*avrB*) challenged *srg1 atgsnor1-3* and *srg1 nox1* mutants. The extent of *S*-nitrosylation of SRG1 was increased in *atgsnor1-3* and *nox1* mutants relative to *srg1* (Fig. 4h). Thus, enhanced

*S*-nitrosylation of SRG1 appears to directly correlate with its loss of transcriptional repression activity.

**Identification of SRG1 *S*-nitrosylation sites.** SRG1 has 7 Cys residues, 4 of which are located within the ZF domain (Fig. 5a). To identify the target site(s) of SRG1 *S*-nitrosylation, the residues outside of the ZF domain, Cys18, Cys28 and Cys143, were mutated either individually or in combination to serine (Ser). The resulting recombinant proteins were subsequently subjected to the BST. Our data indicated that the generated SRG1 single, double and triple mutants were all *S*-nitrosylated following exposure to GSNO (Fig. 5b). Thus, one or more of the four Cys residues within the ZF motif must be target(s) of SNO formation. Utilising a mass spectrometry (MS) approach, we employed a differential labelling strategy in order to covalently modify Cys residues which were susceptible to *S*-nitrosylation with iodoacetamide, thus producing Cys carbamidomethylation (CAM) modification at these positions. The resulting labelled SRG1 was subject to limited trypsin digestion and analysed by Fourier transform ion cyclotron resonance mass spectrometry (FT-ICR MS). Using this approach, 70% sequence coverage of the SRG1 protein was achieved (Supplementary Figure 7), which included 6 of the 7 Cys residues (no information was obtained on Cys143). Similar to our mutational studies, no evidence was observed for *S*-nitrosylation of Cys18, Cys28 and Cys143. However, CAM modification, indicative of *S*-nitrosylation, was observed in the peptide spanning Arg35–Lys60 (containing Cys39 and Cys42) and in the peptide spanning Thr79–Lys111 (containing Cys87 and Cys90) (Supplementary Figure 7). For both of these peptides, peptide masses were observed which contain one and two CAM modifications - suggesting that, for each ZF-TF motif, either (i) both cysteines are susceptible to *S*-nitrosylation; or (ii) *S*-nitrosylation of one cysteine in the ZF-TF motif leads to disulfide bond formation between the two-cysteine ligands in the ZF-TF motif. Collectively, these data suggest that the four Cys residues within the SRG1 ZF motif were potential targets for SNO formation and Cys18, Cys28, and Cys143 may not be susceptible to *S*-nitrosylation.

To confirm and extend these results we explored at which SRG1 Cys residues SNO formation might reduce cognate transcriptional repression. Thus, we determined if mutation of Cys18, Cys28 and Cys143 of SRG1, all of which are located outside the ZF motif, could abolish the ability of SRG1 to function as a transcriptional repressor in vivo. Utilizing an *Arabidopsis* transient transcription activity assay our results indicated that the transcriptional repression activity of the triple Cys18, Cys28 and Cys143 SRG1 mutant was not significantly different from wild-type SRG1 (Fig. 5c). In contrast, SRG1 mediated transcriptional repression was abolished either following addition of the NO donor, SNP (Fig. 5d) or in a *srg1 atgsnor1-3* genetic background (Fig. 5e). These data therefore imply that *S*-nitrosylation of one or more Cys residues within the two ZF-TF motifs might disable the ability of SRG1 to function as a transcriptional repressor.

To reveal the possible impact of NO on the structure/function of the ZF domain of SRG1, we analysed the second ZF domain of this protein by homology modelling[50]. As expected, Cys87 and Cys90 were found to be required to coordinate the $Zn^{2+}$ ion, while neither Cys18, Cys28 or Cys143 were found to function in the coordination of this divalent metal ion. Further, the model suggests that if the thiol group of Cys87 or Cys90 is *S*-nitrosylated, formation of the ZF structure is impaired due to the inability to coordinate the $Zn^{2+}$ ion (Supplementary Figure 8a, 8b and 8c). Therefore, SNO formation at either Cys87 or Cys90 might disrupt formation of the ZF and adjacent EAR domain within SRG1.

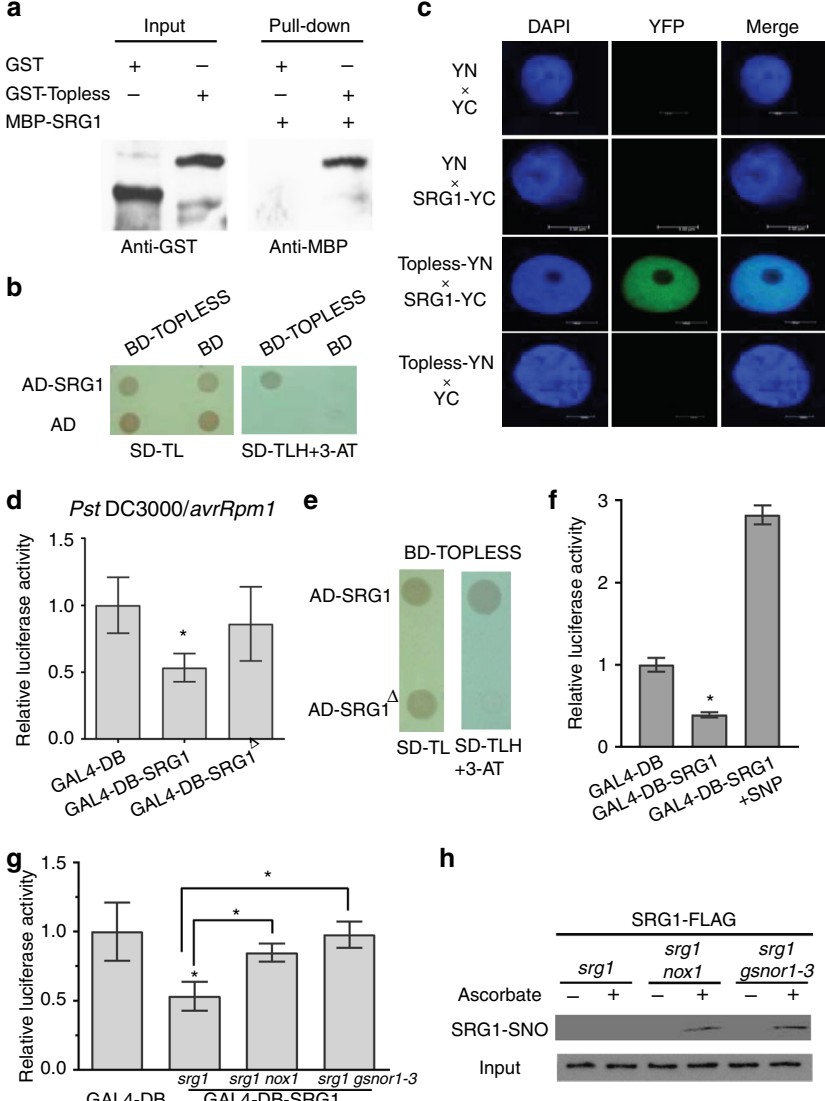

**Fig. 4** NO negatively regulates SRG1 transcriptional repression. **a** Recombinant myelin basic protein (MBP)-SRG1 interacts with glutathione S-transferase (GST) tagged Topless corepressor in an in-vitro pull-down assay, as indicated in western blot analysis utilizing either anti-MBP or anti-GST antibodies. **b** SRG1 interacts with Topless in a yeast two-hybrid assay. AD, Gal4 activation domain; BD, GAL4 DNA-binding domain. Transformed yeast were spotted on selective medium SD-TL or SD-TLH (SD-TLH, synthetic dextrose medium lacking leucine, tryptophan and histidine) with 5 mM 3-amino-1,2,4 triazole (3AT). **c** In planta interaction of SRG1 and Topless established a by bimolecular fluorescence complementation (BiFC) assay in *Nicotiana benthamiana* leaves. Topless was fused to the N-terminus of yellow fluorescent protein (YFP) to form Topless-YN and SRG1 was fused to the C-terminus of YFP forming SRG-YC. These proteins were then simultaneously transiently expressed in *N. benthamiana* leaves by agroinfiltration. *In planta* interaction of Topless-YN and SRG-YC enables YFP-dependent fluorescence imaged by confocal microscopy. Scale bar, 5 µm. **d** Relative luciferase activity was measured after cotransformation of reporter plasmids together with an effector plasmid, comprised of either the GAL4 DNA-binding domain (DB) (GAL4-DB), GAL4-SRG1-DB or GAL4-DB-SRG1△ (SRG1△, deletion of EAR domain of SRG1) into *srg1* protoplasts. Values are mean ± SD ($n = 3$). Star indicates significant difference compared to GAL4-DB at $p < 0.05$ by student t-test. **e** Assay for interaction SRG1 deleted of the EAR motif (SRG1△) with Topless using a yeast two-hybrid assay. **f** Inhibition of SRG1 transcriptional repression activity by the nitric oxide donor, SNP (1 mM), in a transient repression activity assay as previously outlined. Error bars represent ± SD from 3 biological replicates. Asterisks indicate a significant difference compared with GAL4-DB (t-test, *$P < 0.05$). **g** Transient repression activity assay for SRG1 in the stated plant lines. Error bars represent ± SD ($n = 3$, t-test, *$P < 0.05$ compared with GAL4-DB or GAL4-DB-SRG1 in Col-0, respectively). **h** SRG1 *S*-nitrosylation in the stated plant genotypes. Ascorbate was employed as indicated to control for SNO formation and the input indicated that the total SRG1-FLAG was used for IP

**S-nitrosylation negatively regulates SRG1 immune function.** To test this model, we generated a SRG1 Cys87Histidine (His) substitution. Similar to Cys, His can coordinate the $Zn^{2+}$ ion in protein structures[51] and mass spectrometry analysis suggested that Cys87 might constitute a major site of SRG1-SNO formation (Supplementary Figure 7). SRG1 Cys87His complemented the growth phenotype of *srg1* plants and further, chIP analysis of this modified protein indicated a similar binding profile to SRG1 in

the absence of pathogen challenge (Supplementary Figure 9a–c). Subsequently, we explored the biological consequence of this Cys mutation on the ability of SRG1 to function as a transcriptional repressor in vivo in either the presence or absence of NO. Interestingly, abolishing *S*-nitrosylation at SRG1 Cys87 was sufficient to diminish NO-mediated inhibition of SRG1 transcriptional repressor function (Fig. 5f). Consistent with this observation, expression of SRG1 Cys87His in *srg1* plants, again

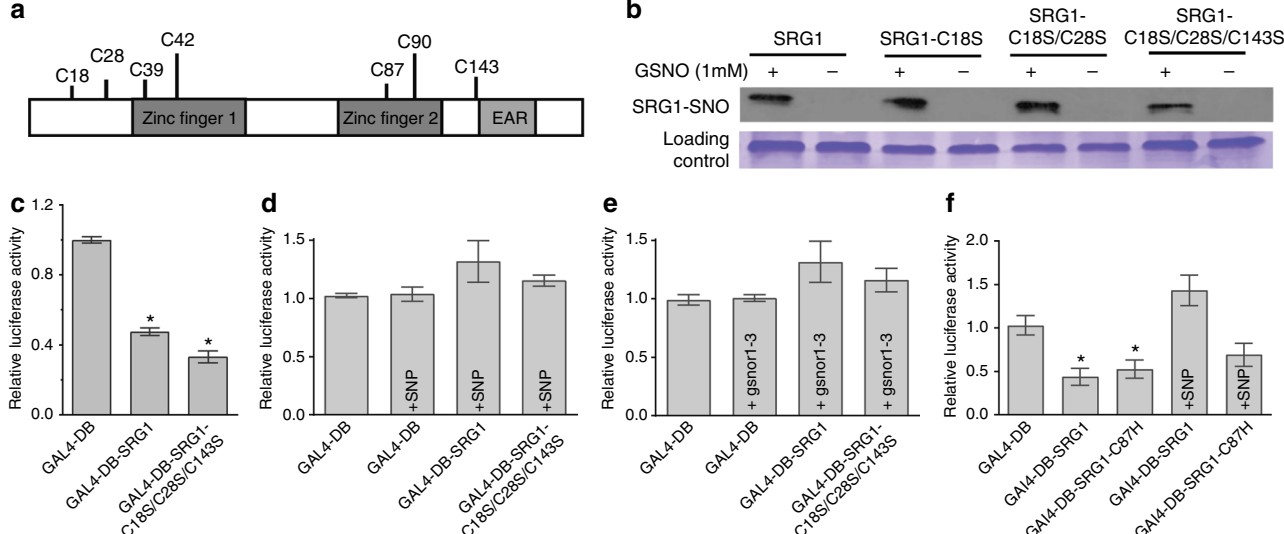

**Fig. 5** *S*-nitrosylation of target SRG1 cysteine residues. **a** Schematic representation of SRG1 showing its predicted functional domain structure and the location of cysteine (Cys) residues. **b** Impact of combined mutations of SRG1 Cys residues outside of the zinc finger (ZF) domains on the *S*-nitrosylation status of SRG1. **c** Impact of combined mutations of SRG1 Cys residues outside of the ZF domain on the transcriptional repression activity of SRG1 in an *srg1* genetic background. Error bars mean ± SD ($n = 3$). Asterisks indicate significant difference relative to GAL4-DB ($t$-test, *$P < 0.05$). **d, e** The impact of the nitric oxide (NO) donor SNP (1 mM) (**d**) and increased *S*-nitrosoglutathione (GSNO) accumulation in a *srg1 gsnor1-3* genetic background (**e**) on the transcriptional repression activity of wild-type (WT) and a triple SRG1 mutant carrying mutations in all Cys residues outside of the ZF domain. **f** Transcriptional repression activity of SRG1 Cys87His relative to wild-type SRG1 in the presence or absence of the NO donor, SNP (1 mM)

precluding SNO formation at this residue, potentiated the immune response, resulting in both increased *Pst* DC3000 (*avrRpm1*)-triggered cell death and decreased bacterial titre (Supplementary Figure 9d-f). Collectively, our data suggests that *S*-nitrosylation of SRG1 at Cys87 serves to negatively regulate SRG1 function, curbing the plant immune response.

**Increased SNO abolish SRG1-dependent immune activation**. To further explore the role of *SRG1* in plant immunity, we crossed the CaMV35S::*SRG1* transgene into a *atgsnor1-3* genetic background, to determine the possible impact of increased SNO levels on phenotypes resulting from *SRG1* overexpression. CaMV35S::*SRG1 atgsnor1-3* plants resembled the *atgsnor1-3* line in terms of stature (Fig. 6a) and fresh weight (Fig. 6b), suggesting CaMV35S::*SRG1* mediated inhibition of *Arabidopsis* growth is repressed by increased SNO levels within a *atgsnor1-3* genetic background. Further, consistent with our data suggesting that binding activity of the SRG1 transcriptional repressor towards sequences present within its own promoter is reduced by NO bioactivity, *SRG1* expression was increased in the CaMV35S::*SRG1 atgsnor1-3* line relative to CaMV35S::*SRG1* wild-type plants. Further, basal *SRG1* transcript accumulation was slightly reduced in *atgsnor1-3* plants which constitutively accumulate SNOs relative to wild-type (Fig. 6c). These results are therefore consistent with our previous data suggesting increasing (S)NO levels appear to suppress SRG1 transcriptional repressor activity.

Strikingly, DAB staining reporting extracellular $H_2O_2$ accumulation was conspicuously reduced in the CaMV35S::*SRG1 atgsnor1-3* line relative to CaMV35S::*SRG1* wild-type plants (Fig. 6d). In a similar fashion, *PR1* expression was also decreased (Fig. 6e). Further, leaf infiltration of *Pst* DC3000 revealed that CaMV35S::*SRG1 atgsnor1-3* plants supported an increased titre of these bacteria relative to the CaMV35S::*SRG1* line (Fig. 6f). In aggregate, these data imply that increased (S)NO levels within an *atgsnor1-3* genetic background abolish CaMV35S::*SRG1*-dependent activation of key immune responses and the associated promotion of basal disease resistance.

## Discussion

While NO is well established as a global regulator of plant defence gene expression[9,10,21,23–26], how this small, mobile signal might function in the nucleus to control the transcription of a plethora of incongruent defence-related genes remains to be established. Our findings suggest a molecular framework for SRG1 activity during plant immune function (Fig. 6g). Following a pathogen-triggered nitrosative burst, transient NO accumulation promotes the expression of SRG1. Subsequently this ZF-TF binds to repeated $AGTN_6AGT$ and $ACTN_4ACT$ motifs within the promoter(s) of target gene(s) which may encode negative regulator(s) of the plant immune response. The EAR domain within SRG1 then recruits the corepressor TOPLESS, suppressing the transcription of the target immune repressor(s), contributing to the activation of plant defences. At later stages of the immune response, as total (S)NO levels rise, the cellular pool of SRG1 becomes increasingly *S*-nitrosylated, with Cys87 a major site of this redox-based modification. SRG1-SNO formation disrupts the coordination of the integral $Zn^{2+}$ ion which may result in a conformational change reducing both cognate DNA binding and transcriptional repression activity. Subsequently, this may enable the expression of one or more immune repressors, which contribute to a negative feedback loop curbing the plant defence response.

In parallel with our findings, in some contexts prolonged NO accumulation has also been demonstrated to display immuno-suppressive activities in mammals[52–55]. Further, inactivation of ZF transcription factors by NO might be a key feature of this immunosuppressive activity. In human cells, NO is thought to abrogate the DNA-binding activities of the ZF-TFs specificity protein 1 (Sp1) and early growth response protein 1 (EGR1), contributing to the repression of interleukin-2 (IL-2) dependent gene expression[56]. Our data implies that SRG1 might function as a nuclear NO sensor-regulator, modulating the transcription of plant defence genes in response to changes in (S)NO concentrations. The Cys residues within the ZF DNA-binding motif of this protein class have long been regarded as possible targets for NO modulation[57], however, there is a paucity of detailed

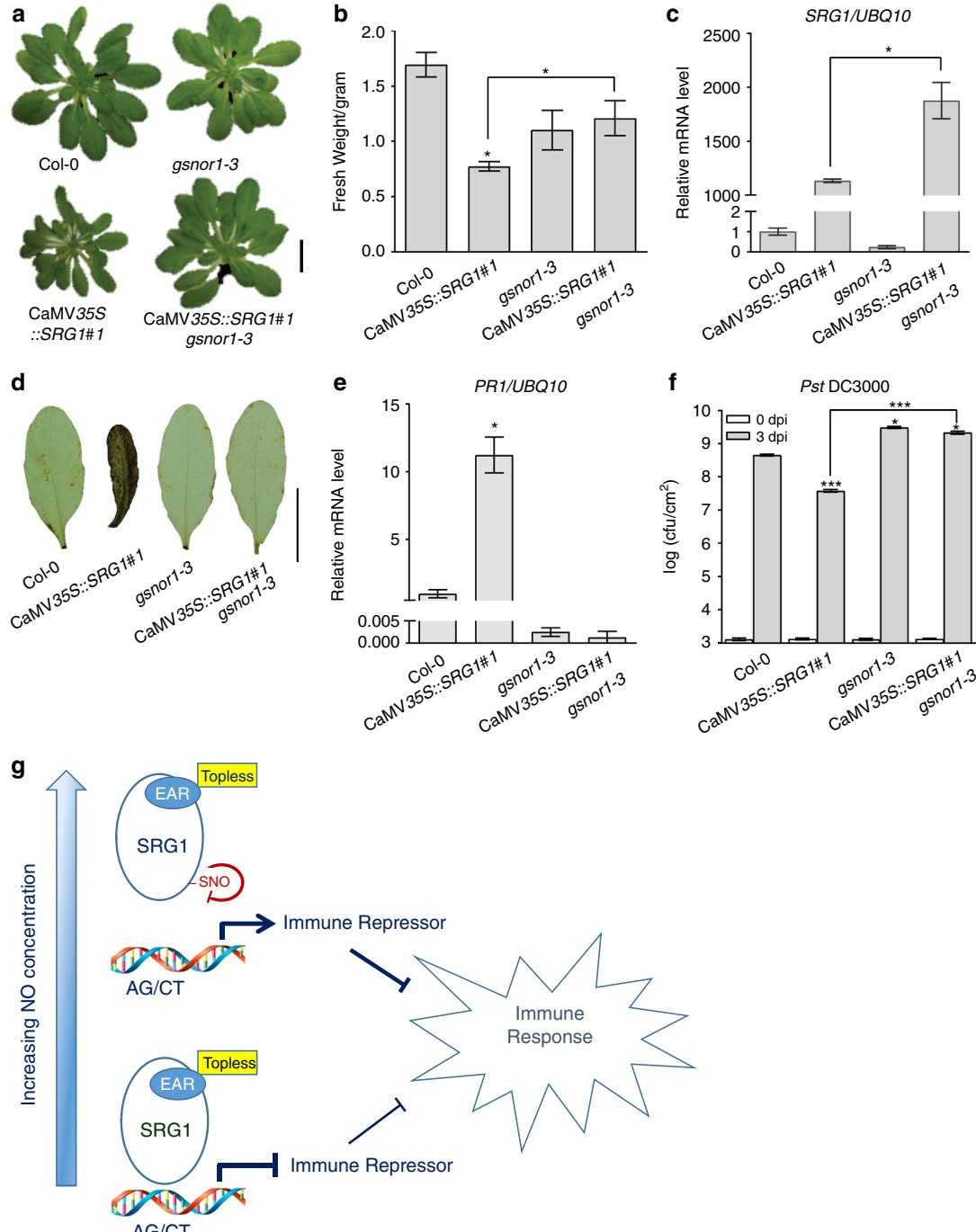

**Fig. 6** SRG1 *S*-nitrosylation negatively regulates immunity. **a** Morphological phenotype of 6-week-old CaMV*35S::SRG1 gsnor1-3* plants in comparison to the stated plant lines. Scale bar = 2 cm. **b**, **c** The fresh weight (**b**) and *SRG1* mRNA level (**c**) of the indicated plant genotypes. **d** Hydrogen peroxide accumulation was determined by 3,3′-diaminobenzidine (DAB) staining in the given lines. Scale bar = 0.5 cm. **e** *PR1* expression analysis determined by qPCR. The mRNA level of Col-0 wild-type plants was normalized as 1.0. Error bars mean ± SD (*n* = 3). Asterisks indicate a significant difference compared with Col-0 (*t*-test, \**P* < 0.05). **f** Titre of *Pst* DC3000 in the given plant genotypes at 0 or 3 days post infiltration. Error bars indicate the mean ± SD from at least three independent biological replicates. Asterisks indicates a significant difference compared to either Col-0 or CaMV*35S::SRG1* with *t*-test, \*\*\**P* < 0.001, \**P* < 0.05. **g** Model for the function of SRG1 in plant immunity. Upon pathogen recognition, nitric oxide (NO) production induces *SRG1* expression. SRG1 contributes to activation of the defence response by suppressing the transcription of one or more immune repressors. As (S)NO levels rise SRG1 becomes *S*-nitrosylated, blunting its DNA binding activity, enabling the expression of one or more immune repressors, priming a negative feedback loop to restrict immune function

molecular insight. Although NO has been shown to drive the release of Zn from the Cys rich metal storage protein, metallothionein[58]. Our model suggests that *S*-nitrosylation of Cys87 and possibly other Cys residues within the separated but paired ZF motifs of SRG1, might result in $Zn^{2+}$ ion release and the concomitant disruption of protein structure, abolishing DNA binding and transcriptional repression. Further, DNA bound ZF-TFs are thought to be significantly less susceptible to NO than the

corresponding unbound proteins[56]. Implying, that the turnover of DNA bound SRG1 may also influence the kinetics of SRG1 inactivation, in addition to the concentration gradient of increasing NO levels.

While either overexpression of SRG1 or the preclusion of SRG1 Cys87 S-nitrosylation promotes cell death development, loss of SRG1 function appears to slow the rate of cellular execution. Thus, redox modulation of this ZF-TF might also control the kinetics of cell death formation. Similarly in mammals, NO inhibits the binding activity of the ZF transcriptional repressor Yin-Yang 1, enabling Fas expression and the subsequent sensitization of cells to Fas cell surface death receptor (Fas)-induced apoptosis[59].

In Saccharomyces cerevisiae, Fzf1p, a C2H2 ZF-TF, has also been shown to drive NO responsive transcription[60]. Further, in Candida albicans, a dimorphic fungus responsible for a considerable proportion of fungal infections in humans, a ZF-TF, CTA4, is also responsible for mediating NO-dependent gene induction. CTA4 belongs to the Zn(II)2-Cys6 transcription factor family, a group of proteins unique to fungi, whose members bind DNA by means of a binuclear cluster of six cysteine residues that coordinate two zinc atoms. In a similar fashion to SRG1, the transcription of CTA4 is induced by NO and deletion of this TF significantly reduced the virulence of C. albicans[61]. The molecular basis underpinning how these fungal ZF-TFs perceive and respond to NO, however, still remains to be established.

Our findings imply that S-nitrosylation of Cys87, an evolutionary conserved residue within the DNA-binding domain of the C2H2 ZF-TF, SRG1, disrupts $Zn^{2+}$ coordination, decreasing the DNA binding and transcriptional repression activity of this TF. Subsequently, this may result in the release from transcriptional suppression of one or more negative regulators of the defence response, curbing the expression of plant immune function. As Cys87 and associated Cys residues within SRG1 are highly conserved among ZF-TFs, this molecular mechanism might underpin the redox regulation of these proteins across phylogeny.

## Methods

**Plant materials and pathogen inoculation**. Arabidopsis Wild-type Col-0, nox1 and gsnor1-3 mutant plants were used for this study. srg1 (SALK_119663) mutant was obtained from NASC. For transgenic plants expressing SRG1, the coding sequence of SRG1 was cloned into the FLAG-containing Gateway vector pGWB11 (Invitrogen) to generate 35 s::SRG1-FLAG. The 1894 bp SRG1 promoter was cloned into the GUS-containing Gateway vector pGWB3 to generate SRG1pro::GUS. Recombinant plasmids were confirmed by sequencing and then transferred into Arabidopsis Col-0 mediated by Agrobacterium tumefaciens GV3101 to generate single copy, homozygous CaMV35S::SRG1 and SRG1Pro::GUS lines, respectively. For srg1-complemented lines, SRG1 genomic sequence with promoter was cloned into destination vector pGWB1 via Gateway system to generate SRG1Pro::SRG1. Plasmid SRG1Pro::SRG1 was transformed into the srg1 line to generate srg1-complemented lines (srg1-C). Transgenic plants were selected on half MS medium containing 50 µg ml⁻¹ kanamycin and homozygous lines with a single insertion were used for experiments.

The srg1 (SALK_119663) mutant was confirmed by PCR based on three primers according to Salk Institute Genomic Analysis Laboratory, with primers (Supplementary Table 1). CaMV35S::SRG1#1 gsnor1-3 was generated by crossing CaMV35S::SRG1#1 with gsnor1-3 plants. Additionally, F2 progeny were screened by PCR with three primers to select gsnor1-3 homozygosity and half MS with kanamycin was used for selecting CaMV35S::SRG1#1 homozygosity. The homozygous CaMV35S::SRG1#1 gsnor1-3 was used for this study. In a similar fashion, gsnor1-3 plants were crossed with srg1 and homozygosity selected as described above for each mutation.

Pst DC3000 was grown in LB medium and inoculation with $5 \times 10^5$ cell by pressure infiltration[62].

**Interrogation of nitric oxide-regulated gene expression**. We interrogated both public and in-house data sets[24,27] for genes that were strongly and rapidly induced by NO. Further, we searched for genes that appeared in all databases reporting (S) NO-induced gene expression. From these genes we prioritised those that were rapidly induced by NO, exhibited a high level of induction by this redox cue and

encoded regulatory proteins. In this fashion, we identified SRG1 and a number of other ZF-TFs that were strongly and rapidly activated by NO.

**Histochemical analysis and confocal microscopy**. The protocol for GUS, DAB and NBT staining were stained and then photographed[62]. Cell death was visualized with trypan blue staining[11]. The transient expression of 35 s::SRG1-GFP in Arabidopsis protoplasts and expression of SRG1-YC and TOPLESS-NC in N. benthamiana were observed under confocal microscopy[62].

**Site-directed mutagenesis**. Site-directed mutagenesis of SRG1 was carried out with QuickChange II Site-Directed Mutagenesis kit (Stratagene).

**Quantitative RT-PCR**. Gene expression was analyzed by qRT-PCR using the calculation with the $2^{-\triangle\triangle Ct}$ with UBQ10 as internal control[62] and gene-specific primers were shown in Supplementary Table 1.

**Recombinant protein expression**. The protein coding sequence of SRG1 was cloned into pDEST-HisMBP in frame and then transformed into E. coli BL21 (DE3). The N-terminal sequence of Topless (1–188 AA) was cloned into pDEST15 to generate GST-Topless. Recombinant MBP-SRG1 was produced in E. coli BL21 (ED3) by adding 0.1 mM IPTG for 4 h and was purified by Amylose Magnetic Beads (NEB, UK). Recombinant protein GST-Topless was produced in E. coli BL21 (ED3) by adding 0.3 mM IPTG for 6 h and was purified by Glutathione Sepharose 4B (GE Healthcare).

**S-nitrosylation assays**. In vitro and in vivo S-nitrosylation assays were conducted by Biotin-switch assay[11,36,63]. Full images of blots were shown in Supplementary Figure 10.

**Transient transcriptional repression activity assay**. The transient transcriptional repression activity assay in Arabidopsis protoplasts was assayed by calculating the relative LUC activity[48]. Full-length SRG1 or the truncated EAR domain of SRG1, SRG1Δ, (Supplementary Figure 5) were amplified and subsequently cloned into effector plasmid GAL4-DB to generate GAL4-DB-SRG1 or GAL4-DB-SRG1Δ (Supplementary Figure 6). Reporter, effector and internal plasmids were co-transformed to stated Arabidopsis protoplasts for 16 h under light and total protein was extracted for luciferase assay according to Dual-luciferase reporter assay kit (Promega).

**Electrophoretic mobility shift assays**. The binding reaction contained 1.9 µg of purified recombinant protein in a buffer with 20 mM HEPES (pH 7.9), 50 mM KCl, 5 mM MgCl₂, 5% glycerol and 0.5 µg ul⁻¹ poly d(I-C). The GSNO was prepared fresh by mixing same amount of 1 M glutathione and 1 M NaNO₂. Dilution were made to add the corresponding volume of 0.25, 1, 10 and 50 mM GSNO to the samples[42]. All procedures were undertaken in a dark room. After 10 min of GSNO addition the DNA labelled with γ³²P-ATP was added. Samples were incubated for 25 min room temperature and then loaded into a gel. A 6% non-denaturing polyacrylamide gel was used to run the reaction with TBE buffer (pH 8.8), 30% acrylamide/bis (29:1), 10% APS and TEMED. The gel was composed with a stacking gel containing the same components except TBE buffer (pH 6.5). The gel was run in darkness for 80 min at 100 V, then dried at 80 °C for 2 h and exposed overnight to an X-ray film with two intensifier screens. Full images of blots were shown in Supplementary Figure 10.

**Frequency of ACTN₄ACT and AGTN₆AGT motifs**. DNA motif frequency in the Arabidopsis genome was determined by Python motif finder.

**Mass spectrometry**. For NO-oxidation, recombinant protein was incubated in HEN buffer (100 µL, 250 mM Hepes-NaOH PH 7.7, 1 mM EDTA and 0.1 mM neocuproine) with the NO donor GSNO (1 mM) and incubated at 25 °C for 20 min in the dark. After incubation, the NO donor was removed by desalting on a Micro Bio-Spin P6 column (BioRad) before alkylation of free cysteines by addition of 300 µL blocking buffer (2.5% SDS and 20 mM NEM in HEN buffer) and incubation at 50 °C for 20 min. Finally, excess NEM was quenched and oxidative modifications removed by addition of 200 mM DTT and subsequent incubation in the dark for 20 min. Samples were then separated by SDS-PAGE on a 4–12% Bis-Tris gel and protein bands excised and subjected to reduction, iodoacetamide (IAM) alkylation and trypsin digestion following standard procedures. Using this workflow, cysteines susceptible to NO oxidation were labelled with NEM (C6H7NO2; Δmass 125.0477 Da) and cysteines unreactive to GSNO were labelled with IAM (C2H3NO; Δmass 57.0215 Da). The resulting peptide mixtures were desalted by C18 reverse phase Zip-Tips (Millipore) before nano-electrospray mass spectrometry (MS).

High resolution MS was performed on a 12T SolariX FT-ICR MS (Bruker Daltonics), with nanospray using a nanomate infusion robot (Advion Boisciences). Resulting peptide mass spectra were calibrated using ESI Tuning Mix (Adilent Technologies) and analysed by Data Analysis software (Bruker Daltonics). A mass

list was created using the SNAP 2.0 algorithm (Bruker Daltonics) and searched against the known protein sequence using MS-Fit software (University of California). For data searching, error tolerances were set to 10 ppm.

**ChIP assay**. ChIP was performed on *Arabidopsis* protoplasts derived from *srg1* or *srg1 gsnor1-3* mutant under short days. The SRG1-GFP was transformed to protoplasts for 16 h and then subjected to ChIP assay with GFP-antibody[43] (ChromoTek, Germany) (Supplementary Table 2). Untreated sonicated chromatin was reverse cross-linked and used as total input DNA for a qRT-PCR experiment. Quantitative PCR using SRG1 promoter specific primers was carried out and *EIF4A1* (*At3g13920*) employed as a control.

**Protein modelling**. SRG1 protein sequence was submitted to Phyre intensive search tool to identify the structural homologues[64,65]. This identified C2H2-type ZF domain transcription factors (Krueppel-like factor 4 (Klf4) as a potential homologue. SNO at Cys87 and Cys90 was built by using MacPyMOL (Version 1.8)[66].

## Data availability
The data sets generated and/or analysed during this study are available from the corresponding author on reasonable request.

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

## Acknowledgements

B.C, Q.P received scholarships from the China Scholarship Council (CSC) and the National Natural Science Foundation of China (grant no. 31700241) and Science and Technology Planning Project of Jiangsu Province of China (SBK2017040656) to Q.P. G.J.L. received funding from BBSRC grant (BB/DO11809/1). D.J.C. would like to acknowledge the University of Edinburgh for a Chancellor's Fellowship. We would like to thank Dr. Chengcai Chu (Institute of Developmental Biology and Genetics, China), Dr. Tomotsugu Koyama (Kyoto University, Japan), Dr. Steven Spoel (The University of Edinburgh, UK) and Dr. Paul Birch (University of Dundee, UK) for sharing vectors. W.S. received funding from the Programme of Introducing Talents of Innovative Discipline to Universities (Project 111) from the State Administration of Foreign Experts Affairs (#B18042).

## Author contributions

B.C, Q.P., D.C., M.O.V., B.Y. and S.U. undertook experiments, while W.S., J.J. and G.J.L. supervised the research. B.C. and G.J.L. wrote the paper with all co-authors providing input.

## Additional information

**Competing interests:** The authors declare no competing interests.

