## [Peer Review File · Nature Communications]

Reviewers' comments:

Reviewer #1 (Remarks to the Author):

S-nitrosylation of the zinc finger protein, SRG1, regulates plant immunity by Cui et al.

This manuscript describes potential new findings in mechanisms underpinning NO-dependent plant response to pathogens. It has been widely reported that NO regulates several plant immune responses although the mechanisms involved are still essentially unknown. Here the authors investigate the molecular mechanisms of this NO function. The manuscript aims to show that the zinc finger transcription factor SRG1 is induced after pathogen attack and by a NO donor. An ERF domain of SRG1 enables it to recruit the co-repressor TOPLESS, thus suppressing gene expression. Additionally, SRG1 is target of S-nitrosylation and this modification affects SRG1-dependent transcriptional suppression.

The findings of this manuscript are potentially very exciting. Not only could they elucidate one of the mechanisms of how NO performs its role in plant defence, they could also provide another of the very few available examples where NO regulates gene expression.

Taking everything into consideration I believe that the authors present convincing evidence to support the notion that SRG1 and its NO-mediated modification is a key component in plant defence. I believe that this paper would be suitable for publication in this journal, providing that the following concerns and queries have been addressed:

1. Could the authors please explain in a little more detail about how they got to the transcription factor SRG1?
2. From the experiments shown mainly in Figure 1 (and Suppl. Fig. 1) the authors conclude that NO induces SGR1 and also that this TF is induced after pathogen attack. However, in order to be able to conclude that the induction of SGR1 in response to a pathogen is NO-dependent, a control with cPTIO is lacking in Figure 1c and Supplementary Figure 1c, as other signalling molecules, such as ROS, are produced together with NO during this response, which may also be involved in gene regulation.
3. Figure 2 shows that SRG1 overexpression induces PR1 expression and SA content, nevertheless it would be nice to see how the *srg1* mutant behaves; is it affected in PR1 expression (SA content) under control and after *Pst avrRpm1* infection? It would be logical to think so, although the *srg1* phenotype in response to the pathogen does not seem to be to striking. Would it be possible that some functional redundancy with other TFs may exist?
4. Does Figure 4h show that the genotypes presented express SRG1-FLAG? If so please add this in the legend and in the labelling of the figure and also what does input mean in this figure? Please explain this in the legend. In Figure 3e, please indicate the genotypes presented are expressing SRG1-FLAG on the label. It would also be nice to complete the legend by explaining that after the BST, proteins were immune-precipitated to facilitate the understanding of the figure. The same applies to Figure 3d, and add *avrRpm1* to the label of the figure. All these figures (Figs. 4h, 3e and d) lack appropriate controls for this method. Therefore, controls in which either ascorbate was omitted from the BST method or the samples exposed to UV are critical to ensure that only S-nitrosylated residues are detected.
5. Figure 5 shows that S-nitrosylation of C87 in SRG1 affects transcriptional repression activity but this data does not rule out the possibility that C90 may have the same effect, for me it is not clear if C87 is more prone to be S-nitrosylated or not than C90. Or if both of them may have the same effect on SRG1 transcriptional repression.
6. It has been shown that SRG1 may bind its own promoter; it is supposed that it may be repressing the expression (auto-repression) as is expressed at a higher level in a high SNOs background (*gsnor1-3*). So, shouldn't the SGR1 be S-nitrosylated before you observe the induction of the gene during the defence response? You comment that you observed S-nitrosylation of the protein 12h post infiltration but you that observed induction of the gene 3h post infiltration (earlier). Does this result suggest that

another mechanism may be involved in SRG1 induction?

Reviewer #2 (Remarks to the Author):

Here, the authors report that S-nitrosylation of the zinc finger protein, SRG1, regulates plant immunity.

The involvement of reactive oxygen or nitrogen species in defense responses and cell death is still an exciting field, and so far many groups and papers have addressed this topic. Most cellular functions are sensitive to redox state. This sensitivity is conferred by specialized redox centers in proteins whose regulation is often viewed as a simple on-off switch, corresponding to reduced and oxidized states.

S-nitrosylation is a ubiquitous redox-related modification of cysteine thiol by nitric oxide(NO, which transduces NO bioactivity. New evidence suggests that the products of S-nitrosylation, play key roles in plant disease/resistance and development. Changes in the levels of various SNOs depend on specific defects in both enzymatic and non-enzymatic mechanisms of nitrosothiol formation, processing and degradation. An understanding of these mechanisms is crucial for the development of an integrated model of NO biology. Thus, the story of Gary Loake and colleagues is certainly worth to be published in a journal such as Nature Comm. However, I would suggest that the authors address the following points:

1. Neither in the title nor in the abstract the authors mention that they work with Arabidopsis. Please note, that plant immunity might look different in monocots or against fungi! Please be a bit more cautious!
2. The authors claim: Following the interrogation of both public and in-house data sets, derived from the profiling of Arabidopsis gene expression in response to (S)NO accumulation, we identified a C2H2 type zinc finger transcription factor....Please, If I do this I end up with hundreds of genes responsive to NO. Could the authors please explain what they did exactly?
3. What else is binding/competing/overlapping with to either ACTN6ACT or ACTN4ACT? Could the authors please give some information about frequency of the binding sites in the genome, preferred class of target genes (resistance?), chromosomes?
4. The authors observe a (mild) phenotype of SRG1 ox or srg1 mutants and claim that 'SRG1 negatively regulates Arabidopsis stature'. That is frankly spoken nonsense. There are thousands of phenotypes that are not directly (!) caused by a targeted gene.
5. Finally, one more point to criticize is an uncomplete in vivo/in planta analyses. For a journal such as Nature Comm it would be good to further characterize SRG1 (mutants) by using complementation with SRG1 constructs modified at different cysteins. Do the authors see a possibility? They have everthing at hand - SRG1 mutants, SRG1 constructs with cys-modifications, their transformation works well. Should be feasible.

Reviewer #3 (Remarks to the Author):

This manuscript reports that SNO modification of a zinc finger transcription factor, SRG1, is critical for NO signaling during plant immunity.

The authors showed that pathogen infection or NO induces expression of SRG1. *srg1* mutant showed susceptibility to a bacterial pathogen *Pseudomonas syringae*, whereas overexpression of SRG1 showed resistance, indicating that SRG1 is a positive regulator of immunity against the bacterial pathogen. SRG1 overexpression associated with increased immune responses such as cell death and SA responses. Then, the authors showed that SRG1 is S-nitrosylated by NO and pathogen infection and that SRG1 binds to a specific DNA sequence, which is abolished by NO, suggesting that S-nitrosylation reduces its DNA binding. They also showed that SRG1 interacts with a key transcriptional co-repressor, TOPLESS and is a transcriptional repressor through its EAR motif. They put several efforts to identify functionally important S-nitrosylation site(s) in SRG1 and suggested that C87 may be important for NO-mediated transcriptional de-repression of SRG1. They speculated that SRG1 might repress expression of negative regulator(s) of immunity thereby positively contributing to immunity and that S-nitrosylation of SRG1 weakens its DNA binding activity, suggesting immune suppression by NO via S-nitrosylation of SRG1.

Since Immunity regulation by NO is not well understood, SRG1 regulation by NO would be highly influential to broad audience. However, this reviewer thinks that there are two major weaknesses in this manuscript.

First, evidence for physiological significance of S-nitrosylation at C87 is unsatisfactory. In the authors' model, SNO modification at C87 triggers release of SRG1 from DNA. Does amino acid exchange at C87 affect DNA binding such as in EMSA and ChIP assays? Also does such version of SRG1 complement the mutant phenotype or not? Such experiments may allow the authors to conclude SNO modification of SRG1 at C87 is an important regulation for NO signaling in plant immunity.

Second, it is completely speculative how SRG1 regulates immunity since the authors did not identify target gene(s) of SRG1. I would suggest ChIP-seq experiments to identify target genes. For this, use of a mutant version of SRG1 at C87 and treatment with/without NO or pathogen infection would also strengthen the authors' conclusion.

In summary, this manuscript is easy to follow and most experiments are solid and would be broadly influential if further evidence is provided.

S-nitrosylation of the zinc finger protein, SRG1, regulates plant immunity
Cui et al. 2018

Referees' comments

Referee #1

1. Could the authors please explain in a little more detail about how they got to the transcription factor SRG1?

We thank this reviewer for highlighting this point. As suggested we have now added some further information on this point (marked in red text) in the Materials and Methods section. We thank the reviewer again for this good suggestion.

2. From the experiments shown mainly in Figure 1 (and Suppl. Fig. 1) the authors conclude that NO induces SGR1 and also that this TF is induced after pathogen attack. However, in order to be able to conclude that the induction of SGR1 in response to a pathogen is NO-dependent, a control with cPTIO is lacking in Figure 1c and Supplementary Figure 1c, as other signalling molecules, such as ROS, are produced together with NO during this response, which may also be involved in gene regulation.

This reviewer raises an excellent point. Following their suggestions we have now added the cPTIO data to Figure 1 and Supplemental Figure 1. This data shows that cPTIO scavenging of nitric oxide (NO) significantly reduces SRG1 transcriptional activation by pathogen challenge. Therefore, this data further substantiates our suggestion that *SRG1* is transcriptionally regulated by NO during pathogen challenge. We thank the reviewer again for this excellent suggestion.

3. Figure 2 shows that SRG1 overexpression induces PR1 expression and SA content, nevertheless it would be nice to see how the *sg1* mutant behaves; is it affected in PR1 expression (SA content) under control and after *Pst avrRpm1* infection?

We thank the reviewer for raising this excellent point. We have now undertaken the suggested experiment and included this information in Supplemental Figure 2e. The expression of *PR1* is indeed reduced in *sg1* plants, as predicted by the reviewer.

Would it be possible that some functional redundancy with other TFs may exist?

This is possible and something we plan to investigate in the future but we feel this is beyond the scope of the current manuscript.

4. Does Figure 4h show that the genotypes presented express SRG1-FLAG? If so please add this in the legend and in the labelling of the figure and also what does input mean in this figure? Please explain this in the legend. In Figure 3e, please indicate the genotypes presented are expressing SRG1-FLAG on the label. It would also be nice to complete the legend by explaining that after the BST, proteins were immune-precipitated to facilitate the understanding of the figure. The same applies to Figure 3d, and add *avrRpm1* to the label of the figure. All these figures (Figs. 4h, 3e and d) lack appropriate controls for this method. Therefore, controls in which either ascorbate was omitted from the BST method or the samples exposed to UV are critical to ensure that only S-nitrosylated residues are detected.

We thank this reviewer for their helpful advice on the labelling of figure 4h, 3d and 3e. As proposed by this reviewer, we have now modified the labelling of these figures and the associated legend accordingly.

As also suggested by this reviewer, we have now added labelling to Figure 4h showing the indicated plant lines express SRG1-FLAG and also amended the text of this Figure legend to highlight this.

We thank this reviewer for pointing out helpful additional controls for Figures 3d, 3e and 4h. The resulting data now provides stronger support to our previous conclusions.

5. Figure 5 shows that S-nitrosylation of C87 in SRG1 affects transcriptional repression activity but this data does not rule out the possibility that C90 may have the same effect, for me it is not clear if C87 is more prone to be S-nitrosylated or not than C90. Or if both of them may have the same effect on SRG1 transcriptional repression.

We do not claim Cys87 is the only Cys to make a contribution to the redox regulation of SRG1. “Our model suggests that S-nitrosylation of Cys87 and possibly other Cys residues within the separated but paired ZF motifs of SRG1, might result in Zn²⁺ ion release and the concomitant disruption of protein structure, abolishing DNA binding and transcriptional repression”. Through *in vitro* and *in vivo* analysis, in addition to protein modelling, we identify Cys87 as a major site of SRG1 redox regulation but it may not be the only site. We feel a detailed structure-function analysis of each potential site of redox-regulation is beyond the scope of the current paper. For example, Tada et al. 2008 Science, showed Cys156 regulated NPR1, but did not exhaustively go on to test the other Cys residues in the 246-N-terminal residues of this transcriptional cofactor. This was despite the fact that NPR1 was already a very well characterized protein. In contrast, our paper has the additional novelty of being the first report of SRG1 function.

6. It has been shown that SRG1 may bind its own promoter; it is supposed that it may be repressing the expression (auto-repression) as is expressed at a higher level in a high SNOs background (gsnor1-3). So, shouldn't the SGR1 be S-nitrosylated before you observe the induction of the gene during the defence response? You comment that you observed S-nitrosylation of the protein 12h post infiltration but you that observed induction of the gene 3h post infiltration (earlier). Does this result suggest that another mechanism may be involved in SRG1 induction?

Yes, there are likely additional mechanisms that contributes to SRG1 induction. As this reviewer will know, eukaryotic gene regulation is highly complex and typically requires an array of transcriptional activators, repressors and chromatin remodeling factors. NO is required to drive a striking induction of SRG1 from very low basal levels. So, in the absence of pathogen challenge and an associated NO burst, the amount of SRG1 and by extension, any associated auto-repression, might be minimal. Therefore, there may be additional NO-mediated mechanisms for the transcriptional activation of SRG1.

Reviewer #2

Here, the authors report that S-nitrosylation of the zinc finger protein, SRG1, regulates plant immunity.

The involvement of reactive oxygen or nitrogen species in defense responses and cell death is still an exciting field, and so far many groups and papers have addressed this topic. Most cellular functions are sensitive to redox state. This sensitivity is conferred by specialized redox centers in proteins whose regulation is often viewed as a simple on-off switch, corresponding to reduced and oxidized states.

S-nitrosylation is a ubiquitous redox-related modification of cysteine thiol by nitric oxide(NO, which transduces NO bioactivity. New evidence suggests that the products of S-nitrosylation, play key roles in plant disease/resistance and development. Changes in the levels of various SNOs depend on specific defects in both enzymatic and non-enzymatic mechanisms of nitrosothiol formation, processing and degradation. An understanding of these mechanisms is crucial for the development of an integrated model of NO biology. Thus, the story of Gary Loake and colleagues is certainly worth to be published in a journal such as Nature Comm. However, I would suggest that the authors address the following points.

We thank this reviewer for their kind support of our manuscript.

1. Neither in the title nor in the abstract the authors mention that they work with *Arabidopsis*. Please note, that plant immunity might look different in monocots or against fungi! Please be a bit more cautious!

The reviewer makes a good point here. We agree. We have added *Arabidopsis* to the abstract to make this clear.

2. The authors claim: Following the interrogation of both public and in-house data sets, derived from the profiling of *Arabidopsis* gene expression in response to (S)NO accumulation, we identified a C2H2 type zinc finger transcription factor....Please, If I do this I end up with hundreds of genes responsive to NO. Could the authors please explain what they did exactly?

We thank this reviewer for highlighting this point. We have now added more explanation on this issue to the Materials and Methods.

3. What else is binding/competing/overlapping with to either ACTN6ACT or ACTN4ACT? Could the authors please give some information about frequency of the binding sites in the genome, preferred class of target genes (resistance?), chromosomes?

Thanks, this reviewer makes another good point. The frequency of the SRG1 binding sites in the *Arabidopsis* genome was calculated by scanning the genome. We have listed the frequency of SRG1 sites on each *Arabidopsis* chromosome, along with the total number of sites in Supplementary Figure 4i and j. Identifying potential target genes is not so straightforward. This subject will be the focus of our next SRG1 paper.

4. The authors observe a (mild) phenotype of SRG1 ox or srg1 mutants and claim that 'SRG1 negatively regulates *Arabidopsis* stature'. That is frankly spoken nonsense. There are thousands of phenotypes that are not directly (!) caused by a targeted gene.

Yes, we agree with this reviewer. We don't claim direct regulation. We have changed this sentence to "SRG1 function negatively impacts *Arabidopsis* stature" to increase the clarity here.

5. Finally, one more point to criticize is an uncomplete in vivo/in planta analyses. For a journal such as Nature Comm it would be good to further characterize SRG1 (mutants) by using complementation with SRG1 constructs modified at different cysteins. Do the authors see a possibility? They have everthing at hand - SRG1 mutants, SRG1 constructs with cys-modifications, their transformation works well. Should be feasible.

We thank this reviewer for their excellent suggestion. Reviewer 3 also requested us to show that SRG1 Cys87His can complement a *srg1* mutant line. As advised, we generated transgenic *Arabidopsis* lines expressing SRG1 C87H in a *srg1* genetic background. This

transgene complements the mutant phenotype of the *srg1* line (Supplementary Figure 9a) and we quantified this (Supplementary Figure 9b). Further, we carried out chIP analysis of this modified transcription factor, which exhibited similar binding to its cognate *cis*-element *in planta* in the absence of pathogen challenge (Supplementary Figure 9c). As suggested by reviewer 3, these experiments now strengthen our evidence suggesting that the SNO modification of SRG1 at C87 maybe an important feature of NO signalling during plant immunity. We thank reviewer 2 again for their suggestion.

Reviewer #3

Since Immunity regulation by NO is not well understood, SRG1 regulation by NO would be highly influential to broad audience.

We thank this reviewer for their kind support of our manuscript.

First, evidence for physiological significance of S-nitrosylation at C87 is unsatisfactory. In the authors' model, SNO modification at C87 triggers release of SRG1 from DNA. Does amino acid exchange at C87 affect DNA binding such as in EMSA and ChIP assays? Also does such version of SRG1 complement the mutant phenotype or not? Such experiments may allow the authors to conclude SNO modification of SRG1 at C87 is an important regulation for NO signaling in plant immunity.

We thank this reviewer for their excellent suggestions. As advised, we generated transgenic lines expressing SRG1 C87H in a *srg1* genetic background. This transgene complements the mutant phenotype of the *srg1* line (Supplementary Fig.9a and b). Further, we carried out chIP analysis of this modified transcription factor, which exhibited similar binding to its cognate *cis*-element *in planta* in the absence of pathogen challenge (Supplementary Fig.9c). As described by this reviewer, these experiments now further enable us to conclude that SNO modification of SRG1 at C87 maybe an important feature of NO signalling during plant immunity. We thank this reviewer again for their excellent suggestions.

Second, it is completely speculative how SRG1 regulates immunity since the authors did not identify target gene(s) of SRG1. I would suggest ChIP-seq experiments to identify target genes. For this, use of a mutant version of SRG1 at C87 and treatment with/without NO or pathogen infection would also strengthen the authors' conclusion.

This is a good suggestion by this reviewer but we feel this experimental work is beyond the scope and focus of this first SRG1 paper. The proposed work executed carefully represents a significant amount of time, effort and funding. For example, the ChIP-seq analysis of the transcription factor SARD1 was published relatively recently as a stand-alone paper in Nature Communications (Sun et al. Nature Comm 2015). The identification of SRG1 target genes will hopefully be the subject of our next SRG1 paper.

In summary, this manuscript is very well written, easy to follow and most experiments are solid and would be broadly influential if further evidence is provided.

We thank this reviewer for their support and encouragement.

Reviewers' comments:

Reviewer #1 (Remarks to the Author):

The Manuscript has been improved and the suggestion of this referee has been taken into account. As I mentioned before, the findings of this manuscript are very exciting and provide a mechanism of how NO performs its role in plant defence, and how NO regulates gene expression.

Reviewer #2 (Remarks to the Author):

The paper improved substantially

Reviewer #3 (Remarks to the Author):

Although the current manuscript has been improved from the original version, this reviewer thinks that the authors did not address the two major points that I raised on the previous version.

First, regarding the physiological significance of S-nitrosylation at C87 (or at least the residue itself) of SRG1, the authors added FigS9 to show SRG1 C87H is not different from the wild-type sequence for SRG1 function. To this reviewer, this figure shows that C87 is not important for the function of SRG1, and therefore these new data do not support the authors' conclusions. The last 8 lines of Abstract are about Cys87 of SRG1. Therefore, I think the authors should show the relevance of S-nitrosylation at C87 (or at least the residue itself) in physiological processes. Since SRG1 C87H in *srg1* is available, the authors can do experiments such as in FigS2b-e using this material. If Cys87 is relevant for the release of immune repressors from negative regulation by SRG1, SRG1 C87H in *srg1* should behave as *srg1* in immunity (more bacterial growth and less PR1 expression compared to WT SRG1 in *srg1*).

Second, the authors argue that identification of SRG1 target genes is beyond of their scope. I still think that this would much strengthen the quality of this manuscript. However, since this issue is not related to presented contents of this manuscript, I would leave this to the editor.

NCOMMS-17-23316

**S-nitrosylation of the zinc finger protein, SRG1, regulates plant immunity
Cui et al. 2018**

Referees' comments

Referee #3

Although the current manuscript has been improved from the original version, this reviewer thinks that the authors did not address the two major points that I raised on the previous version.

First, regarding the physiological significance of S-nitrosylation at C87 (or at least the residue itself) of SRG1, the authors added FigS9 to show SRG1 C87H is not different from the wild-type sequence for SRG1 function. To this reviewer, this figure shows that C87 is not important for the function of SRG1, and therefore these new data do not support the authors' conclusions. The last 8 lines of Abstract are about Cys87 of SRG1. Therefore, I think the authors should show the relevance of S-nitrosylation at C87 (or at least the residue itself) in physiological processes. Since SRG1 C87H in *srg1* is available, the authors can do experiments such as in FigS2b-e using this material. If Cys87 is relevant for the release of immune repressors from negative regulation by SRG1, SRG1 C87H in *srg1* should behave as *srg1* in immunity (more bacterial growth and less PR1 expression compared to WT SRG1 in *srg1*).

As suggested by this reviewer, we have now undertaken additional experiments to explore the potential function of S-nitrosylation at Cys87 in SRG1 function during the plant immune response.

SRG1 acts as a positive regulator of immunity, by repressing the transcription of one or more immune suppressors. S-nitrosylation at Cys87 in SRG1, abolishes SRG1 DNA binding, thereby enabling the expression of one or more immune repressors, curbing the immune response. Thus, preventing the S-nitrosylation of SRG1 at Cys87 would be expected to extend the transcriptional repression of one or more immune suppressors, resulting in an enhanced immune response.

Consistent with this posit, our data shows that SRG1 Cys87His mutant lines potentiate both pathogen-triggered cell death and the restriction of bacterial titre. This data highlights a key role for the regulation of SRG1 function in plant immunity through the S-nitrosylation of Cys87.

Second, the authors argue that identification of SRG1 target genes is beyond of their scope. I still think that this would much strengthen the quality of this manuscript. However, since this issue is not related to presented contents of this manuscript, I would leave this to the editor.

We appreciate the suggestion by this reviewer of identifying SRG1 target genes. However, the editor has not requested this addition work, which we feel is beyond the scope and focus of this first SRG1 paper. Therefore, we have not included an additional ChIP-seq based study in the current manuscript.

We thank this reviewer for their very helpful advice enabling us to further improve our paper.

REVIEWERS' COMMENTS:

Reviewer #3 (Remarks to the Author):

The authors have satisfactorily addressed my previous comments. I think that this work significantly advances our understanding of the role of NO and S-nitrosylation in plant defense.